# LISA: Layerwise Importance Sampling for Memory-Efficient Large Language Model Fine-Tuning

**Rui Pan♠\***,  **Xiang Liu♣\***,  **Shizhe Diao♦**,  **Renjie Pi♡**,  **Jipeng Zhang♡**,
**Chi Han♠**,  **Tong Zhang♠**

♠University of Illinois Urbana-Champaign
♣The Hong Kong University of Science and Technology(Guangzhou)
♦NVIDIA  ♡The Hong Kong University of Science and Technology
{ruip4, chihan3, tozhang}@illinois.edu
xliu886@connect.hkust-gz.edu.cn   {sdiaoaa, rpi, jzhanggr}@ust.hk

## Abstract

The machine learning community has witnessed impressive advancements since large language models (LLMs) first appeared. Yet, their massive memory consumption has become a significant roadblock to large-scale training. For instance, a 7B model typically requires at least 60 GB of GPU memory with full parameter training, which presents challenges for researchers without access to high-resource environments. Parameter efficient fine-tuning techniques such as Low-Rank Adaptation (LoRA) have been proposed to alleviate this problem. However, in most large-scale fine-tuning settings, their performance does not reach the level of full parameter training because they confine the parameter search to a low-rank subspace. Attempting to complement this deficiency, we investigate the layerwise properties of LoRA on fine-tuning tasks and observe an unexpected but consistent skewness of weight norms across different layers. Utilizing this key observation, a surprisingly simple training strategy is discovered, which outperforms both LoRA and full parameter training in a wide range of settings with memory costs as low as LoRA. We name it **L**ayerwise **I**mportance **S**ampled **A**damW (**LISA**), a promising alternative for LoRA, which applies the idea of importance sampling to different layers in LLMs and randomly freeze most middle layers during optimization. Experimental results show that with similar or less GPU memory consumption, LISA surpasses LoRA or even full parameter tuning in downstream fine-tuning tasks, where LISA consistently outperforms LoRA by over 10%-35% in terms of MT-Bench score while achieving on-par or better performance in MMLU, AGIEval and WinoGrande. On large models, specifically LLaMA-2-70B, LISA surpasses LoRA on MT-Bench, GSM8K, and PubMedQA, demonstrating its effectiveness across different domains.

## 1  Introduction

Large language models (LLMs) like ChatGPT excel in tasks such as writing documents, generating complex code, answering questions, and conducting human-like conversations [1]. With LLMs being increasingly applied in diverse task domains, domain-specific fine-tuning has emerged as a critical strategy to enhance their downstream capabilities [2, 3, 4, 5]. Nevertheless, these methods are typically time-intensive and consume substantial computational resources, posing significant challenges to the development of large-scale models [6]. For example, continual pre-training typically

---

*Equal Contribution.

38th Conference on Neural Information Processing Systems (NeurIPS 2024).

requires several weeks even with multiple 80 GB GPUs. To reduce costs, Parameter-Efficient Fine-Tuning (PEFT) techniques have been proposed to minimize the number of trainable parameters.

These techniques include adapter weights [7], prompt weights [8], and LoRA [9]. Among these, LoRA stands out as one of the most widely adopted due to its unique ability to merge the adaptor back into the base model parameters, significantly enhancing efficiency. However, LoRA's superior performance in fine-tuning tasks has yet to reach a point that universally surpasses full parameter fine-tuning in all settings [10, 11]. In particular, it has been observed that LoRA tends to falter on large-scale datasets during continual pre-training [12], which raises doubts about the effectiveness of LoRA under those circumstances. We attribute this to LoRA's much fewer trainable parameters compared to the base model, which limits the representation power of LoRA training.

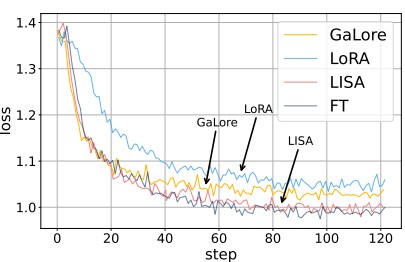

Figure 1: Training loss of LLaMA-2-7B on Alpaca GPT-4.

To overcome this shortcoming, we delve into LoRA's training statistics in each layer, aspiring to bridge the difference between LoRA and full-parameter fine-tuning. Surprisingly, we discover that LoRA's layerwise weight norms have an uncommonly skewed distribution, where the bottom layer and/or the top layer occupy the majority of weights during the update. In contrast, the other self-attention layers only account for a small amount, which means different layers have different importance when updating. This key observation inspires us to "sample" different layers by their importance, which matches the idea of importance sampling [13, 14].

As a natural consequence, this strategy brings forth our **L**ayerwise **I**mportance **S**ampled **A**dam (**LISA**) algorithm, where by selectively updating only essential LLM layers and leaving others untouched, LISA enables training large-scale language models ($\geq$ 65B parameters) with less or similar memory consumption as LoRA. Furthermore, fine-tuned on downstream tasks, LISA outperformed both LoRA and conventional full-parameter fine-tuning approaches by a large margin, indicating the large potential of LISA as a promising alternative to LoRA.

We summarize our key contributions as follows,

- We discover the phenomenon of skewed weight-norm distribution across layers in LoRA, which implies the varied importance of different layers in large-scale LLM training.

- We propose the Layerwise Importance Sampled AdamW (LISA), a simple optimization method capable of scaling up to over 70B LLMs with less or similar memory cost as LoRA.

- We demonstrate LISA's effectiveness in fine-tuning tasks for modern LLMs, where it outperforms LoRA by 10%-35% in MT-Bench and achieves better performance in multiple benchmarks. In addition, LISA exhibits much better convergence behaviors than LoRA. LISA even outperforms full parameters training under certain settings. Similar performance gain is observed across different sized models (7B-70B) and tasks, including instruction following, medical QA, and math problems.

## 2 Related Work

### 2.1 Large Language Models

In the realm of natural language processing (NLP), the Transformer architecture has been a revolutionary technique, initially known for its effectiveness in machine translation tasks [15]. With the inception of models like BERT [16] and GPT-2 [17], the approach shifted towards pre-training on extensive corpora, which led to significant performance enhancements in downstream fine-tuning tasks [2, 18, 19, 20, 21, 22, 23, 24, 25, 26, 27]. However, the growing number of parameters in these models results in a huge GPU memory consumption, rendering the fine-tuning of large scale models ($\geq$ 65B) infeasible under low resource scenarios. This has prompted a shift towards more efficient training of LLMs.

## 2.2 Parameter-Effieient Fine-Tuning

Parameter-efficient fine-tuning (PEFT) methods adapt pre-trained models by fine-tuning only a subset of parameters. In general, PEFT methods can be grouped into three classes: 1) Prompt Learning methods [8, 28, 29, 30, 31, 32, 33], 2) Adapter methods [7, 9, 34, 35, 36, 37, 38], and 3) Selective methods [39, 39, 40, 41]. Prompt learning methods emphasize optimizing the input token or input embedding with frozen model parameters, which generally has the least training cost among all three types. Adapter methods normally introduce an auxiliary module with much fewer parameters than the original model, and updates are only applied to the adapter module during training. Compared with them, selective methods are more closely related to LISA, which focuses on optimizing a fraction of the model's parameters without appending extra modules. Recent advances in this domain have introduced several notable techniques through layer freezing. AutoFreeze [39] offers an adaptive mechanism to identify layers for freezing automatically and accelerates the training process. FreezeOut [42] progressively freezes intermediate layers, significantly reducing training time without notably affecting accuracy. The SmartFRZ [40] framework utilizes an attention-based predictor for layer selection, substantially cutting computation and training time while maintaining accuracy. However, none of these layer-freezing strategies has been widely adopted in the context of Large Language Models due to their inherent complexity or non-compatibility with modern memory reduction techniques [43, 44, 27] for LLMs.

## 2.3 Low-Rank Adaptation (LoRA)

In contrast, the Low-Rank Adaptation (LoRA) technique is much more prevalent in common LLM training [9]. LoRA reduces the number of trainable parameters by employing low-rank matrices, thereby lessening the computational burden and memory cost. One key strength of LoRA is its compatibility with models featuring linear layers, where the decomposed low-rank matrices can be merged back into the original model. This allows for efficient deployment without changing the model architecture. As a result, LoRA can be seamlessly combined with other techniques, such as quantization [11] or Mixture of Experts [45]. Despite these advantages, LoRA's performance is not universally comparable with full parameter fine-tuning. There have been tasks in [10] that LoRA performs much worse than full parameter training on. This phenomenon is especially evident in large-scale pre-training settings [12], where to the best of our knowledge, only full parameter training was adopted for successful open-source LLMs [21, 22, 23, 46, 47, 26, 27].

## 2.4 Large-scale Optimization Algorithms

In addition to approaches that change model architectures, there have also been efforts to improve the efficiency of optimization algorithms for LLMs. One such approach is layerwise optimization, a concept with roots extending back several decades. Notably, [48] introduced an effective layer-by-layer pre-training method for Deep Belief Networks (DBN), demonstrating the benefits of sequential layer optimization. This idea was expanded by researchers like [49, 50], who illustrated the advantages of a greedy, unsupervised approach to pre-training each layer of deep networks. In the context of large batch training, [51, 52] developed LARS and LAMB to improve generalization and mitigate the performance declines associated with large batch sizes. Despite these innovations, Adam [53, 54, 55, 27] and AdamW [56] continue to be the predominant optimization methods used in most LLM settings.

Recently, other attempts have also been made to reduce the training cost of LLMs. For example, MeZO [57] adopted zeroth order optimization, bringing significant memory savings during training. However, it also incurred a considerable performance drop in multiple benchmarks, particularly in complex fine-tuning scenarios. Regarding acceleration, Sophia [58] incorporates clipped second-order information into the optimization, obtaining non-trivial speedup on LLM training. The significant downsides are its intrinsic complexity of Hessian estimation and unverified empirical performance in large-size models (e.g., $\geq 65B$). In parallel to our work, [59] proposed GaLore, a memory-efficient training strategy that reduces memory cost by projecting gradients into a low-rank compact space. Yet the performance has still not surpassed full-parameter training in fine-tuning settings. To sum up, LoRA-variant methods [9, 11, 59] with AdamW [56] is still the dominant paradigm for large-size LLM fine-tuning, the performance of which still demands further improvements.

## 3 Method

### 3.1 Motivation

To understand how LoRA achieves effective training with only a few parameters, we conducted empirical studies on multiple models, especially observing the weight norms across various layers. We fine-tune it on the Alpaca-GPT4 dataset [60]. During the training, we meticulously recorded the mean weight norms of each layer $\ell$ at every step $t$ after updates, i.e.

$$\mathbf{w}^{(\ell)} \triangleq \texttt{mean-weight-norm}(\ell) = \frac{1}{T} \sum_{t=1}^{T} \|\boldsymbol{\theta}_t^{(\ell)}\|_2$$

Figure 2 presents these findings, with the x-axis representing the layer id, from embedding weights to the final layer, and the y-axis quantifying the weight norm. The visualization reveals one key trend:

- The embedding layer or the language modeling (LM) head layer exhibits significantly larger weight norms than intermediary layers in LoRA, often by a factor of hundreds. This phenomenon, however, was not salient under full-parameter training settings.

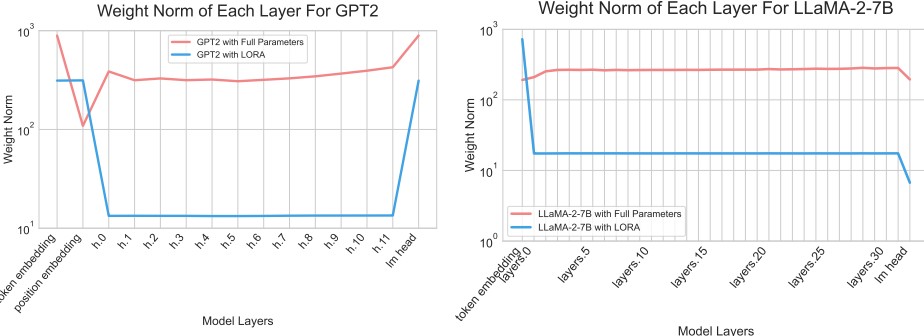

Figure 2: Layer-wise weight norms during training of GPT2 and LLaMA-2-7B Model with LoRA and Full Parameters training.

This observation indicates that the update emphasis of LoRA and full parameter training differ significantly, which can be attributed to the difference in their learned knowledge. For example, in embedding layers, tokens with similar meanings, i.e., synonyms, can be projected into the same embedding space and converted to similar embeddings. LoRA may capture this similarity in language and "group" them in the low-dimension space, allowing frequent features of language meanings to be promptly identified and optimized. The price is LoRA's limited representation power restricted by its intrinsic low-rank space, as we can see from the comparison with LISA in image generation tasks (Appendix A.1), where LoRA memorizes and learns details much slower than LISA. Other possible explanations can also justify this phenomenon. Despite various interpretations of this observation, one fact remains clear: *LoRA values layerwise importance differently from full parameter tuning*.

### 3.2 Layerwise Importance Sampled AdamW (LISA)

To exploit the discovery above, we aspire to simulate LoRA's updating pattern via sampling different layers to freeze. This way, we can avoid LoRA's inherent deficiency of limited low-rank representation ability and emulate its fast learning process. Intuitively, given the same global learning rates across layers, layers with small weight norms in LoRA should also have small sampling probabilities to unfreeze in full-parameter settings so the expected learning rates across iterations can stay the same. This is exactly the idea of importance sampling [13, 14], where instead of applying layerwise different learning rates $\{\eta_t\}$ in full-parameter settings to emulate LoRA's updates $\{\tilde{\eta}_t\}$, we apply sampling and instead get the same expected parameter update

$$\eta_t^{(\ell)} = \tilde{\eta}_t^{(\ell)} \cdot \frac{\tilde{\mathbf{w}}^{(\ell)}}{\mathbf{w}^{(\ell)}} \quad \Rightarrow \quad \eta_t^{(\ell)} = \eta^{(\ell)}, p^{(\ell)} = \frac{\tilde{\mathbf{w}}^{(\ell)}}{\mathbf{w}^{(\ell)}}$$

This gives rise to our Layerwise Importance Sampling AdamW method, as illustrated in Algorithm 1. In practice, since all layers except the bottom and top layer have small weight norms in LoRA, we adopt $\{p_\ell\}_{\ell=1}^{N_L} = \{1.0, \gamma/N_L, \gamma/N_L, \ldots, \gamma/N_L, 1.0\}$ in practice, where $\gamma$ controls the expected number of unfreeze layers during optimization, and the embedding layer $E$ and head layer $H$ remain active. Intuitively, $\gamma$ serves as a compensation factor to bridge the difference between LoRA and full parameter tuning, letting LISA emulate a similar layerwise update pattern as LoRA. To further control the memory consumption in practical settings, we instead randomly sample $\gamma$ layers every time to upper-bound the maximum number of unfrozen layers during training.

---

**Algorithm 1 L**ayerwise **I**mportance **S**ampling **A**damW (**LISA**)

---

**Require:** number of layers $N_L$, number of iterations $T$, sampling period $K$, number of sampled layers $\gamma$, initial learning rate $\eta_0$
1: **for** $i \leftarrow 0$ to $T/K - 1$ **do**
2:     Freeze all layers except the embedding and language modeling head layer
3:     Randomly sample $\gamma$ intermediate layers to unfreeze
4:     Run AdamW for $K$ iterations with $\{\eta_t\}_{t=ik}^{ik+k-1}$
5: **end for**

---

## 4 Experimental Results

Table 1: The chart illustrates peak GPU memory consumption for various model architectures and configurations, highlighting differences across models. The LISA configuration is specifically labeled in the table: "E" denotes the embedding layer, "H" represents the language modeling head layer, and "2L" indicates two additional intermediate layers. *: Model parallelism is applied for the 70B model.

| | VANILLA | LoRA RANK | | | LISA ACTIVATE LAYERS | | |
| --- | --- | --- | --- | --- | --- | --- | --- |
| MODEL | - | 128 | 256 | 512 | E+H | E+H+2L | E+H+4L |
| GPT2-SMALL | 3.8G | 3.3G | 3.5G | 3.7G | 3.3G | 3.3G | 3.4G |
| TINYLLAMA | 13G | 7.9G | 8.6G | 10G | 7.4G | 8.0G | 8.3G |
| MISTRAL-7B | 59G | 23G | 26G | 28G | 21G | 23G | 24G |
| LLAMA-2-7B | 59G | 23G | 26G | 28G | 21G | 23G | 24G |
| LLAMA-2-70B* | OOM | 79G | OOM | OOM | 71G | 75G | 79G |

### 4.1 Memory Efficiency

We conducted peak GPU memory experiments to demonstrate LISA's memory efficiency and showcase its comparable or lower memory cost than LoRA.

**Settings** To reasonably estimate the memory cost, we randomly sample prompts from the Alpaca dataset [61] and limit the maximum output token length to 1024. We focus on two key hyperparameters: LoRA's rank and LISA's number of activation layers. For other hyperparameters, a mini-batch size of 1 was consistently used across five LLMs from 120M to 70B parameters, deliberately excluding other GPU memory-saving techniques such as gradient checkpointing [62], offloading [63], and flash attention [64, 65]. All memory-efficiency experiments are conducted on $4\times$ NVIDIA Ampere Architecture GPUs with 80G memory.

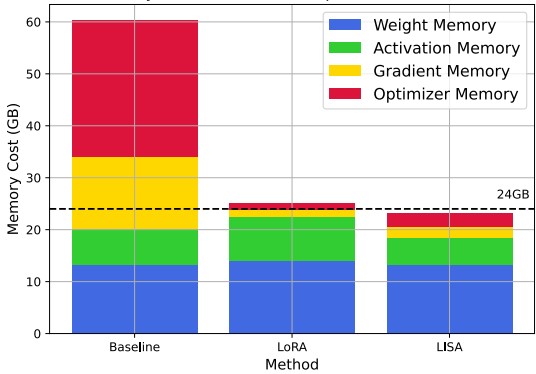

Figure 3: GPU memory consumption of LLaMA-2-7B with different methods and batch size 1.

**Results** Upon examining Table 1, it is evident that the LISA configuration, particularly when enhanced with both the embedding layer (E) and two additional layers (E+H+2L), demonstrates a considerable reduction in GPU memory usage when fine-tuning the LLaMA-2-70B model, as compared to the LoRA method. Specifically, the LISA E+H+2L configuration shows a decrease to 75G of peak GPU memory from the 79G required by the

LoRA Rank 128 configuration. This efficiency gain is not an isolated incident; a systematic memory usage decrease is observed across various model architectures, suggesting that LISA's method of activating layers is inherently more memory-efficient.

In Figure 3, it is worth noticing that the memory reduction in LISA allows LLaMA-2-7B to be trained on a single RTX4090 (24GB) GPU, which makes high-quality fine-tuning affordable even on a laptop computer. In particular, LISA requires much less activation memory consumption than LoRA since it does not introduce additional parameters brought by the adaptor. LISA's activation memory is even slightly less than full parameter training since pytorch [66] with deepspeed [44] allows deletion of redundant activations before backpropagation.

On top of that, a reduction in memory footprint from LISA also leads to an acceleration in speed. As shown in Figure 4, LISA provides almost $2.9\times$ speedup when compared with full-parameter training, and $\sim 1.5\times$ speedup against LoRA, partially due to the removal of adaptor structures. It is worth noticing that the reduction of memory footprint in both LoRA and LISA leads to a significant acceleration of forward propagation, emphasizing the importance of memory-efficient training.

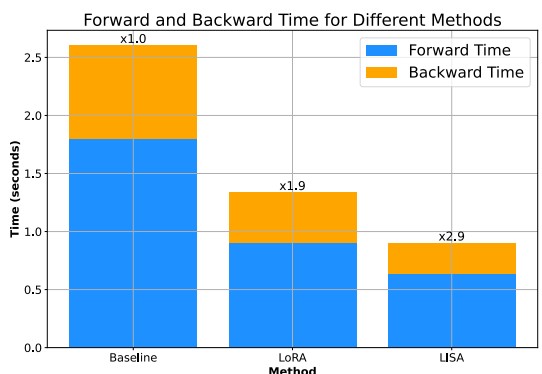

Figure 4: Single-iteration time cost of LLaMA-2-7B with different methods and batch size 1.

## 4.2 Moderate Scale Fine-Tuning

LISA can achieve this significant memory saving while still obtaining competitive performance under the fine-tuning setting.

Table 2: Results of different methods on MMLU, AGIEval, and WinoGrande, measured by accuracy.

| MODEL | METHOD | MMLU (5-SHOT) | AGIEval (3-SHOT) | WINOGRANDE (5-SHOT) |
|---|---|---|---|---|
| TINYLLAMA | VANILLA | 25.50 | 19.55 | 59.91 |
|  | LORA | 25.81 ± 0.07 | 19.82 ± 0.11 | 61.33 ± 0.09 |
|  | GALORE | 25.21 ± 0.06 | 21.19 ± 0.07 | 61.09 ± 0.12 |
|  | **LISA** | **26.02 ± 0.13** | **21.71 ± 0.09** | 61.48 ± 0.08 |
|  | FT | 25.62 ± 0.10 | 21.28 ± 0.07 | **62.12 ± 0.15** |
| MISTRAL-7B | VANILLA | 60.12 | 26.79 | 79.24 |
|  | LORA | 61.78 ± 0.09 | 27.56 ± 0.07 | 78.85 ± 0.11 |
|  | GALORE | 57.87 ± 0.08 | 26.23 ± 0.05 | 75.85 ± 0.13 |
|  | **LISA** | **62.09 ± 0.10** | **29.76 ± 0.09** | **78.93 ± 0.08** |
|  | FT | 61.70 ± 0.13 | 28.07 ± 0.12 | 78.85 ± 0.12 |
| LLAMA-2-7B | VANILLA | 45.87 | 25.69 | 74.11 |
|  | LORA | 45.50 ± 0.07 | 24.73 ±0.04 | 74.74 ± 0.09 |
|  | GALORE | 45.56 ± 0.05 | 24.39 ± 0.11 | 73.32 ± 0.12 |
|  | **LISA** | **46.21 ± 0.12** | 26.06 ± 0.08 | **75.30 ± 0.11** |
|  | FT | 45.66 ± 0.09 | **27.02 ± 0.10** | 75.06 ± 0.13 |

**Settings** To demonstrate the superiority of LISA over LoRA, we evaluate them on the instruction-following fine-tuning task with the Alpaca GPT-4 dataset [61], which consists of 52k conversation pairs generated by GPT-4 [5]. The effectiveness of fine-tuning was evaluated on multiple benchmarks: MT-Bench [67] features 80 high-quality, multi-turn questions designed to assess LLMs on multiple aspects; MMLU [68] includes a total of 57 tasks with 14,079 questions covering a broad spectrum of world knowledge; AGIEval [69] serves as a human-centric benchmark for general abilities, comprising 9,316 instances; WinoGrande [70] is a large-scale dataset for commonsense reasoning, consisting of 44,000 instances designed to challenge models' understanding of the context and commonsense knowledge.

In our experiments, we assessed three baseline models: TinyLlama [47], Mistral-7B [46], and LLaMA-2-7B [23]. These models, varying in size ranging from 1B to 7B parameters, provide a

diverse representation of decoder-only models. For hyper-parameters, we adopt a rank of 128 for LoRA and E+H+2L for LISA in this section, with full details available in Appendix B.

**Results** Table 2 and 3 present a detailed comparison on moderate-scale LLMs. The baselines include Full-parameter Training (FT), Low-Rank Adaptation (LoRA) [9] and Gradient Low-Rank Projection (GaLore) [59]. The results demonstrate that LISA consistently outperforms other fine-tuning methods in most evaluation tracks, indicating its robustness and effectiveness across diverse tasks and model architectures. LISA is particularly effective in instruction following tasks, where a large gap is observed when compared with other baseline methods. LISA even outperforms Full-parameter Training, suggesting that an implicit regularization effect is present when the number of unfrozen layers is restricted, which is similar to dropout [71]. According to more results in stable diffusion and detailed MT-Bench scores, we found that LISA outperforms LoRA mostly in memorization tasks, such as depicting high-resolution image details in image generation, or Writing or Humanities tasks

Table 3: Different methods on MT-Bench.

| MODEL | METHOD | MT-BENCH ↑ |
|---|---|---|
| TINYLLAMA | VANILLA | 1.25 |
| | LORA | 1.90 ± 0.14 |
| | GALORE | **2.61 ± 0.17** |
| | **LISA** | 2.57 ± 0.25 |
| | FT | 2.21 ± 0.16 |
| MISTRAL-7B | VANILLA | 4.32 |
| | LORA | 4.41 ± 0.09 |
| | GALORE | 4.36 ± 0.16 |
| | **LISA** | **4.85 ± 0.14** |
| | FT | 4.64 ± 0.12 |
| LLAMA-2-7B | VANILLA | 3.29 |
| | LORA | 4.45 ± 0.15 |
| | GALORE | 4.63 ± 0.09 |
| | **LISA** | **4.94 ± 0.14** |
| | FT | 4.75 ± 0.16 |

in instruction following. This implies that LISA's performance improvement may majorly come from the ability to memorize long-tailed patterns, while LoRA is better at multi-hop reasoning with limited knowledge. For more details, please refer to Appendix A.1 and A.2.

## 4.3 Moderate Scale Continual Pre-training

Continual pre-training is crucial for enabling models to adapt to new data and domains. To evaluate LISA's efficacy in the continual pre-training scenario, we experiment on the mathematics domain in comparison with Full-parameter Training.

**Settings** We adopt the mathematics corpus OpenWebMath [72] for constructing the continual pre-training dataset. Specifically, we extracted a high-quality subset from it which contains 1.5 billion tokens. Full details are explained in Appendix B.2. After continual pre-

Table 4: Comparison of Moderate Scale Model Continual Pre-training on OpenWebMath Dataset.

| MODEL | METHOD | GSM8K ↑ | MEM. ↓ |
|---|---|---|---|
| TINYLLAMA | VANILLA | 2.26 | - |
| | **LISA** | **3.56** | **8G** |
| | FT | 3.26 | 13G |
| LLAMA-2-7B | VANILLA | 14.40 | - |
| | **LISA** | **22.21** | **26G** |
| | FT | 22.21 | 59G |

trainig, we then apply the same fine-tuning procedure on the GSM8K [73] training set, which comprises 7473 instances.

**Results** Table 4 shows that LISA is capable of achieving on-par or even better performance than full-parameter training with much less memory consumption. Specifically, LISA requires only half of the memory cost compared to full-parameter training. This indicates a better balance between computational efficiency and model performance is achieved by LISA. According to our experience, reducing the number of unfrozen layers to half the original size leads to no worse or even better performance during continual pretraining, while requiring much less memory consumption.

## 4.4 Large Scale Fine-Tuning

To further demonstrate LISA's scalability on large-sized LLMs, we conduct additional fine-tuning experiments on LLaMA-2-70B [23].

**Settings** On top of the aforementioned instruction-following tasks in Section 4.2, we use extra domain-specific fine-tuning tasks on mathematics and medical QA benchmarks. The GSM8K dataset [73], comprising 7473 training instances and 1319 test instances, is used for the mathematics

domain. For the medical domain, we select the PubMedQA dataset [74], which includes 211.3K artificially generated QA training instances and 1K test instances.

Evaluation on the PubMedQA dataset [74] is conducted in a 5-shot prompt setting, while the GSM8K dataset [73] assessment was conducted using Chain-of-Thought (CoT) prompting, following recent studies [75, 76, 77]. Regarding hyperparameters, as detailed in the section 4.1, we utilize the rank 256 for LoRA and the configuration E+H+4L for LISA. Further information is available in Appendix B.

Table 5: Different methods on MT-Bench, GSM8K, and PubMedQA score for LLaMA-2-70B.

| METHOD | MT-BENCH↑ | GSM8K↑ | PUBMEDQA↑ |
|--------|-----------|--------|-----------|
| VANILLA | 5.19 | 54.8 | 83.0 |
| LoRA | 6.10 | 59.4 | 90.8 |
| **LISA** | **6.72** | 61.1 | **91.6** |
| FT | 6.25 | **67.1** | 90.8 |

**Results**   As shown in Table 5, LISA consistently produces better or on-par performance when compared with LoRA. Furthermore, LISA again surpasses full-parameter training in instruction-tuning tasks, providing strong evidence to support LISA's scalability under large-scale training scenarios. More results are available in Appendix A.2.

## 4.5   Ablation Studies

**Hyperparameters of LISA**   The two key hyperparameters of LISA are the number of sampling layers $\gamma$ and sampling period $K$. To obtain intuitive and empirical guidance of those hyperparameter choices, we conduct ablation studies using TinyLlama [47] and LLaMA-2-7B [23] models with the Alpaca-GPT4 dataset. The configurations for $\gamma$, such as E+H+2L, E+H+8L, were denoted as $\gamma = 2$ and $\gamma = 8$. As for the sampling period $K = T/n$, $T = 122$ representing the maximum training step within our experimental framework. The findings, presented in Table 6, reveal that both $\gamma$ and $K$ markedly affect the LISA algorithm's performance. Specifically, a higher $\gamma$ value increases the quantity of trainable parameters, albeit with higher memory costs. On the other hand, an optimal $K$ value facilitates more frequent layer switching, thereby improving performance to a certain threshold, beyond which the performance may deteriorate. Generally, the rule of thumb is: *More sampling layers and higher sampling period lead to better performance.*   For a detailed examination of loss curves and MT-Bench results, refer to Appendix A.4.

Table 6: Different LISA hyperparameters combinations. All settings adopt learning rate $\eta_0 = 10^{-5}$. Here $\gamma$ stands for sampling layers, $K$ stands for sampling period.

| MODELS | $\gamma$ | $K$ | MT-BENCH SCORE |
|--------|----------|-----|----------------|
| TINYLLAMA | 2 | $\lceil T/125 \rceil$ | 2.44 |
| | | $\lceil T/25 \rceil$ | **2.73** |
| | | $\lceil T/5 \rceil$ | 2.64 |
| | | $T$ | 2.26 |
| | 8 | $\lceil T/125 \rceil$ | 2.59 |
| | | $\lceil T/25 \rceil$ | **2.81** |
| | | $\lceil T/5 \rceil$ | 2.74 |
| | | $T$ | 2.53 |
| LLaMA-2-7B | 2 | $\lceil T/125 \rceil$ | 4.86 |
| | | $\lceil T/25 \rceil$ | **4.91** |
| | | $\lceil T/5 \rceil$ | 4.88 |
| | | $T$ | 4.64 |
| | 8 | $\lceil T/125 \rceil$ | 4.94 |
| | | $\lceil T/25 \rceil$ | **5.11** |
| | | $\lceil T/5 \rceil$ | 5.01 |
| | | $T$ | 4.73 |

**Sensitiveness of LISA**   As LISA is algorithmically dependent on the sampling sequence of layers, it is intriguing to see how stable LISA's performance is under the effect of randomness. For this purpose, we further investigate LISA's performance variance over three distinct runs, each with a different random seed for layer selection. Here, we adopt TinyLlama, LLaMA-2-7B, and Mistral-7B models with the Alpaca-GPT4 dataset while keeping all other hyperparameters consistent with those used in the instruction following experiments in section 4.2. As shown in Table 7, LISA is quite resilient to different random seeds, where the performance gap across three runs is within 0.13, a small value compared to the

Table 7: The MT-Bench scores derived from varying random seeds for layer selection.

| MODEL | SEED 1 | SEED 2 | SEED 3 |
|-------|--------|--------|--------|
| TINYLLAMA | 2.57 | 2.55 | 2.60 |
| MISTRAL-7B | 4.85 | 4.82 | 4.82 |
| LLaMA-2-7B | 4.94 | 4.92 | 4.89 |

performance gains over baseline methods. For more ablation experiment on LISA hyperparameters, please refer to Appendix A.4.

## 5  Discussion

**Theoretical Properties of LISA**   Compared with LoRA, which introduces additional parameters and leads to changes in loss objectives, layerwise importance sampling methods enjoy nice convergence guarantees in the original loss. For layerwise importance sampled SGD, similar to gradient sparsification [78, 55], the convergence can still be guaranteed for unbiased estimation of gradients with increased variance. The convergence behavior can be further improved by reducing the variance with appropriately defined importance sampling strategy [14]. For layerwise importance sampled Adam, theoretical results in [79, 55] prove its convergence in convex objectives. If we denote $f$ as the loss function and assume that the stochastic gradients are bounded, then based on [56], we know that AdamW optimizing $f$ aligns with Adam optimizing $f$ with a scaled regularizer, which can be written as

$$f^{\mathrm{reg}}(\mathbf{w}) \triangleq f(\mathbf{w}) + \frac{1}{2}\mathbf{w}^{\top}\mathbf{S}\mathbf{w},$$

where $\mathbf{S}$ is a finite positive semidefinite diagonal matrix. Following existing convergence results of RBC-Adam (Corollary 1 in [79]), we have the convergence guarantee of LISA in Theorem 1.

**Theorem 1** *Let the loss function $f$ be convex and smooth. If the algorithm runs in a bounded convex set and the stochastic gradients are bounded, the sequence $\{\mathbf{w}_t\}_{t=1}^{T}$ generated by LISA admits the following convergence rate:*

$$\frac{1}{T}\sum_{t=1}^{T} f^{\mathrm{reg}}(\mathbf{w}_t) - f_*^{\mathrm{reg}} \leq \mathcal{O}\left(\frac{1}{\sqrt{T}}\right),$$

*where $f_*^{\mathrm{reg}}$ denotes the optimum value of $f^{\mathrm{reg}}$.*

**Memorization and Reasoning**   In our instruction following experiments in Appendix A.1 and A.2, we observe that LISA is much better than LoRA at memorization-centered tasks, such as Writing or depicting image details, while this gap is much smaller in reasoning-centered tasks like Code or Math. It is an intriguing observation since LISA emphasizes more on layer-wise width and restricts the depth of learned parameters, while LoRA focuses more on depth and restricts the representation space in each layer. It may suggest that width is crucial for memorization, while depth is important for reasoning, a similar phenomenon that echos the intuition of [80]. Based on the same intuition, it may be possible to combine the benefits of both and bring forth an even better PEFT method.

## 6  Conclusion

In this paper, we propose Layerwise Importance Sampled AdamW (LISA), an optimization algorithm that randomly freezes layers of LLM based on a given probability. Inspired by observations of LoRA's skewed weight norm distribution, a simple and memory-efficient freezing paradigm is introduced for LLM training. This paradigm achieves significant performance improvements over LoRA on downstream fine-tuning tasks with various models, including LLaMA-2-70B. Further experiments on domain-specific training also demonstrate its effectiveness, showing LISA's huge potential as a promising alternative to LoRA for LLM training.

## Limitations

The major bottleneck of LISA is the same as LoRA, where during optimization, the forward pass still requires the model to be presented in the memory, leading to significant memory consumption. This limitation shall be compensated by approaches similar to QLoRA [11], where we intend to conduct further experiments to verify its performance.

In addition, as suggested by the theoretical intuition, the strategy of E+H+2L in Section 4.2 and E+H+4L in Section 4.4 may not be the optimal importance sampling strategy, given it still sampled

intermediate layers in a uniformly random fashion. We anticipate the optimizer's efficiency will be further improved when considering data sources and model architecture in the importance sampling procedure.

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

# A  Additional Experiments

## A.1  Image Generation

Stable Diffusion has emerged as a powerful approach for generating high-quality images by leveraging the diffusion process in a latent space [81]. This method involves diffusing the data distribution over time and iteratively refining the generated samples to enhance their realism and fidelity. The key innovation lies in operating within the latent space of a pre-trained autoencoder, significantly reducing computational complexity while maintaining high-quality outputs. The Latent Consistency Model (LCM) further improves the efficiency and performance of diffusion models in the latent space by incorporating a consistency loss, which penalizes deviations from the expected latent trajectories [82]. This ensures that the generated samples remain coherent and visually plausible throughout the diffusion steps.

In our experiments, we evaluated the performance of LCM against other fine-tuning methods such as LoRA, and LISA using latent consistency distillation to distill stable diffusion v1.5 and v2.1 model. We utilized the official code[*] and parameters for both LISA and LoRA from the Diffusers library to ensure consistency and comparability of results. Adjustments were made only to the batch size and accumulation steps to achieve a balanced trade-off between computational efficiency and performance. Figure 5 is the generated image comparison. Observations reveal that LISA can generate higher-quality images in fewer inference steps. The images generated using LISA display more intricate details and sharper clarity, particularly evident in the distinct facial features and environment textures. In contrast, the LoRA-generated images offer a softer, more blended aesthetic with a dream-like quality, emphasizing smooth transitions over precise detail. The prompts for images in figure 5 in the left-to-right are given below

- Self-portrait oil painting, a beautiful cyborg with golden hair, 8k.

- Astronaut in a jungle, cold color palette, muted colors, detailed, 8k.

- A photo of a beautiful mountain with a realistic sunset and blue lake, a highly detailed masterpiece.

## A.2  Instruction Following Fine-tuning

Table 8 offers a comprehensive evaluation of three fine-tuning methods—Full Parameter Fine-Tuning (FT), Low-Rank Adaptation (LoRA), Gradient Low-Rank Projection(GaLore), and Layerwise Importance Sampling AdamW (LISA)—across a diverse set of tasks including Writing, Roleplay, Reasoning, Math, Extraction, STEM, and Humanities within the MT-Bench benchmark. The results demonstrate LISA's superior performance, which surpasses LoRA, GaLore, and full parameter tuning in most settings. Notably, LISA consistently outperforms LoRA and full parameter tuning in domains such as Writing, STEM, and Humanities. This implies that LISA can benefit memorization tasks, while LoRA partially favors reasoning tasks.

Table 8: Comparison of Language Model Fine-Tuning Methods on the MT-Bench score.

| MODEL & METHOD | WRITING | ROLEPLAY | REASONING | CODE | MATH | EXTRACTION | STEM | HUMANITIES | AVG. ↑ |
|---|---|---|---|---|---|---|---|---|---|
| TINYLLAMA (VANILLA) | 1.05 | 2.25 | 1.25 | 1.00 | 1.00 | 1.00 | 1.45 | 1.00 | 1.25 |
| TINYLLAMA (LoRA) | 2.77 | 4.05 | 1.35 | 1.00 | **1.40** | 1.00 | 1.55 | 2.15 | 1.90 |
| TINYLLAMA (GALORE) | **3.55** | **5.20** | 2.40 | **1.15** | 1.40 | **1.85** | 2.95 | 2.40 | **2.61** |
| TINYLLAMA (**LISA**) | 3.30 | 4.40 | **2.65** | 1.12 | 1.30 | 1.75 | **3.00** | **3.05** | 2.57 |
| TINYLLAMA (FT) | 3.27 | 3.95 | 1.35 | 1.04 | 1.33 | 1.73 | 2.69 | 2.35 | 2.21 |
| MISTRAL-7B (VANILLA) | 5.25 | 3.20 | 4.50 | 1.60 | 2.70 | **6.50** | **6.17** | 4.65 | 4.32 |
| MISTRAL-7B (LoRA) | 5.30 | 4.40 | 4.65 | **2.35** | **3.30** | 5.50 | 5.55 | 4.30 | 4.41 |
| MISTRAL-7B (GALORE) | 5.05 | **5.27** | 4.45 | 1.70 | 2.50 | 5.21 | 5.52 | 5.20 | 4.36 |
| MISTRAL-7B (**LISA**) | **6.84** | 3.65 | **5.45** | 2.20 | 2.75 | 5.65 | 5.95 | **6.35** | **4.85** |
| MISTRAL-7B (FT) | 5.50 | 4.45 | 5.45 | 2.50 | 3.25 | 5.78 | 4.75 | 5.45 | 4.64 |
| LLAMA-2-7B (VANILLA) | 2.75 | 4.40 | 2.80 | 1.55 | 1.80 | 3.20 | 5.25 | 4.60 | 3.29 |
| LLAMA-2-7B (LoRA) | 6.30 | 5.65 | **4.05** | 1.60 | 1.45 | 4.17 | 6.20 | 6.20 | 4.45 |
| LLAMA-2-7B (GALORE) | 5.60 | 6.40 | 3.20 | 1.25 | 1.95 | **5.05** | 6.57 | 7.00 | 4.63 |
| LLAMA-2-7B (**LISA**) | **6.55** | **6.90** | 3.45 | **1.60** | **2.16** | 4.50 | **6.75** | **7.65** | **4.94** |
| LLAMA-2-7B (FT) | 5.55 | 6.45 | 3.60 | 1.75 | 2.00 | 4.70 | 6.45 | 7.50 | 4.75 |

---

[*]https://github.com/huggingface/diffusers/tree/main/examples/consistency_distillation

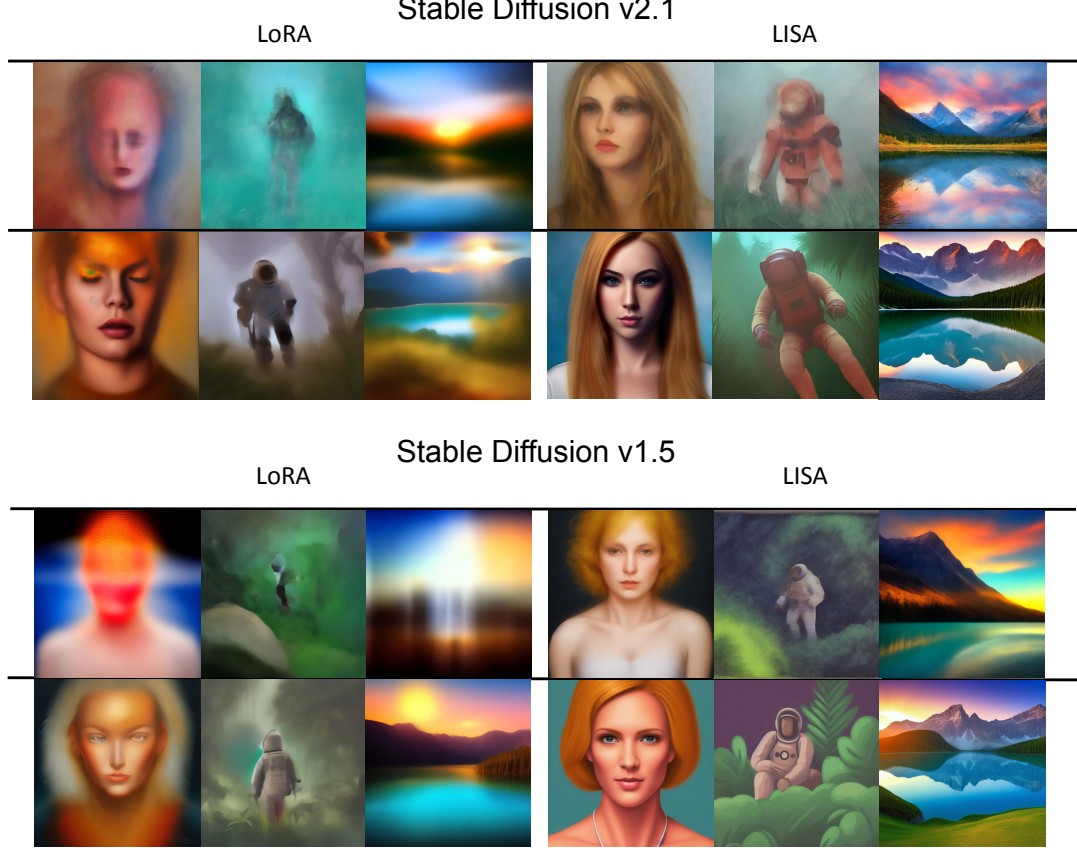

Figure 5: Generated images using LoRA (left) and LISA (right) on Stable Diffusion v2.1 model and Stable Diffusion v1.5. **First row**: number of inference step = 2. **Second row**: number of inference step = 10.

Table 9 provides detailed MT-Bench scores for the LLaMA-2-70B model discussed in Section 4.4, demonstrating LISA's superior performance over LoRA in all aspects under large-scale training scenarios. Furthermore, in Figure 6, we observe that LISA consistently exhibits on-par or faster convergence speed than LoRA across different models, which provides strong evidence for LISA's superiority in practice.

Table 9: Mean score of three fine-tuning methods over three seeds for LLaMA-2-70B on the MT-Bench.

| MODEL & METHOD | WRITING | ROLEPLAY | REASONING | CODE | MATH | EXTRACTION | STEM | HUMANITIES | AVG. ↑ |
|---|---|---|---|---|---|---|---|---|---|
| LLAMA-2-70B(VANILLA) | 7.77 | 5.52 | 2.95 | 1.70 | 1.70 | 6.40 | 7.42 | 8.07 | 5.19 |
| LLAMA-2-70B(LORA) | 7.55 | 7.00 | 5.30 | 3.15 | 2.60 | 6.55 | 8.00 | 8.70 | 6.10 |
| LLAMA-2-70B(**LISA**) | **8.18** | **7.90** | **5.45** | **4.45** | **2.75** | **7.45** | **8.60** | **9.05** | **6.72** |
| LLAMA-2-70B(FT) | 6.45 | 7.50 | 5.50 | 3.40 | 2.15 | 7.55 | 8.10 | 9.40 | 6.25 |

The aforementioned results show that Vanilla LLaMA-2-70B excels in Writing, but full-parameter fine-tuning led to a decline in these areas, a phenomenon known as the "Alignment Tax" [1]. This tax highlights the trade-offs between performance and human alignment in instruction tuning. LISA, however, maintains strong performance across various domains with a lower "Alignment Tax".

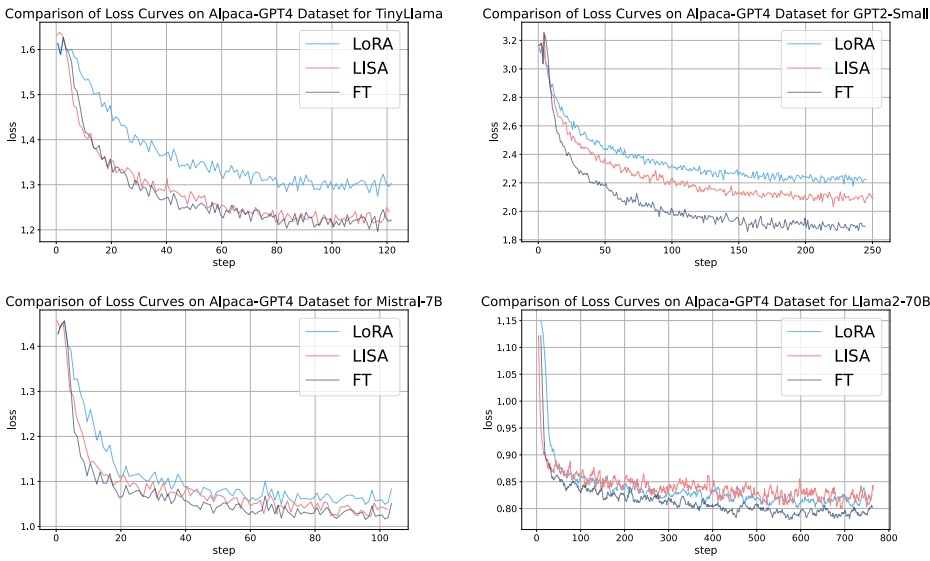

Figure 6: Loss curves for LoRA, LISA, and full-parameter training on the Alpaca-GPT4 dataset across different models.

## A.3 Continual Pre-training

To better analyze the performance of LISA in the continual pre-training scenario, we used OpenWebMath for continual pre-training and the GSM8K train split for the fine-tuning stage, varying the number of sampling layers, $\gamma$, within LISA, ranging from $2, 4, 8, 16$, compared to the accuracy of the full parameter (FT) continual pre-training. Table 7 details the results for the LLaMA-2-7B model under various continual pre-training configurations. Notably, LISA with eight sampling layers achieves comparable accuracy with full parameters continual pre-training method. Furthermore, LISA with 16 sampling layers passes the accuracy of full parameter training.

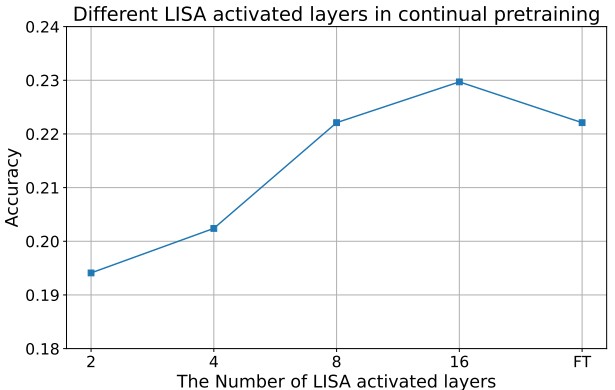

Figure 7: The comparison of full parameter (FT) training and LISA with different sampling layers under continual pre-training scenario. The accuracy is the test set of GSM8K.

## A.4 Ablation Experiments

### A.4.1 Sampling Layers $\gamma$

We conducted an ablation study on the LLaMA-2-7B model trained with the Alpaca-GPT4 dataset, setting the sampling period $K = 13$, so the number of samplings is exactly 10. The study explored different configurations of sampling layers $\gamma$ including {E+H+2L, E+H+4L, E+H+8L}. Figure 8 depicts the impact of the number of sampling layers $\gamma$ on the training dynamics of the model. Three scenarios were analyzed: $\gamma = 2$ (blue line), $\gamma = 4$ (green line), and $\gamma = 8$ (red line), throughout 120 training steps. Initially, all three configurations exhibit a steep decrease in loss, signaling rapid initial improvements in model performance. it's clear that the scenario with $\gamma = 8$ consistently maintains a lower loss compared to the $\gamma = 2$ and $\gamma = 4$ configurations, suggesting that a higher $\gamma$ value leads to better performance in this context.

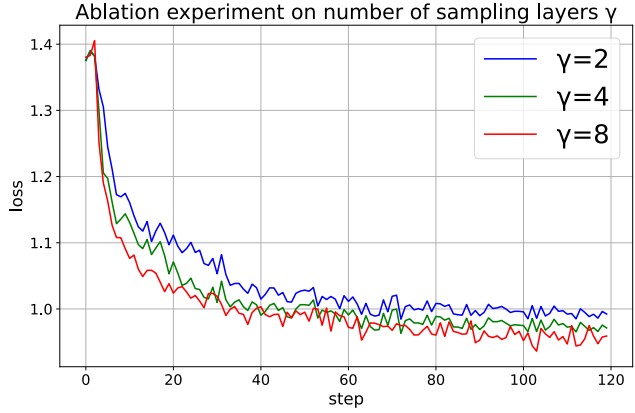

Figure 8: Comparison of loss curves for the $\gamma$ ablation experiment.

Table 10 demonstrates that LISA with only $E + H$ layers experiences a performance decline when the 0 sampling layers $\gamma$ are omitted, highlighting the importance of these layers.

Table 10: Results of additional experiments on LISA with only $E + H$ layers.

| MODEL | METHOD | MMLU (5-SHOT) | AGIEVAL (3-SHOT) | WINOGRANDE (5-SHOT) | MT-BENCH |
|---|---|---|---|---|---|
| TINYLLAMA | VANILLA | 25.50 | 19.55 | 59.91 | 1.25 |
| | LISA (E + H) | 25.49 ± 0.14 | 20.75 ± 0.21 | 60.43 ± 0.19 | 2.18 ± 0.31 |
| | **LISA** | **26.02 ± 0.13** | **21.71 ± 0.09** | 61.48 ± 0.08 | **2.57 ± 0.25** |
| | FT | 25.62 ± 0.10 | 21.28 ± 0.07 | **62.12 ± 0.15** | 2.21 ± 0.16 |
| MISTRAL-7B | VANILLA | 60.12 | 26.79 | 79.24 | 4.32 |
| | LISA (E + H) | 61.49 ± 0.12 | 27.66 ± 0.07 | 77.93 ± 0.11 | 4.51 ± 0.27 |
| | **LISA** | **62.09 ± 0.10** | **29.76 ± 0.09** | **78.93 ± 0.08** | **4.85 ± 0.14** |
| | FT | 61.70 ± 0.13 | 28.07 ± 0.12 | 78.85 ± 0.12 | 4.64 ± 0.12 |
| LLAMA-2-7B | VANILLA | 45.87 | 25.69 | 74.11 | 3.29 |
| | LISA (E + H) | 45.88 ± 0.12 | 25.82 ± 0.15 | 73.48 ± 0.22 | 4.63 ± 0.35 |
| | **LISA** | **46.21 ± 0.12** | 26.06 ± 0.08 | **75.30 ± 0.11** | **4.94 ± 0.14** |
| | FT | 45.66 ± 0.09 | **27.02 ± 0.10** | 75.06 ± 0.13 | 4.75 ± 0.16 |

To better understand the LISA and hyperparameter Sampling Layers $\gamma$, we conducted ablation experiments on Sampling Layers $\gamma$ and learning rate $\eta$. The aim was to investigate the combined impact of these two variables on the LISA. Our experiments utilized the LLaMA-2-7B model, trained on the GSM8K dataset. We examined the effect of increasing the number of Sampling Layers $\gamma$ while simultaneously decreasing the learning rate $\eta$.

Table 11: Compare LISA with different Sampling Layers $\gamma$ and Learning Rate $\eta$, evaluate on GSM8K.

| SAMPLING LAYERS $\gamma$ | LEARNING RATE $\eta$ | | | |
|---|---|---|---|---|
| | $5 \times 10^{-5}$ | $2.5 \times 10^{-5}$ | $1.25 \times 10^{-5}$ | $6.25 \times 10^{-6}$ |
| 2 | **15.77** | 15.32 | 15.21 | 15.01 |
| 4 | 11.29 | 11.34 | **15.87** | 15.27 |
| 8 | 13.32 | 14.39 | **16.30** | 15.32 |
| 16 | 15.42 | 15.92 | 15.78 | **16.57** |

The Table 11, indicates that a higher number of sampling layers can enhance the model's effectiveness, provided the learning rate is adjusted appropriately. Specifically, the optimal performance is observed when the learning rate $\eta$ is reduced in proportion to the increase in the number of sampling layers $\gamma$, demonstrating the delicate balance required between these two parameters to maximize LISA's efficacy.

### A.4.2 Sampling Period $K$

Figure 9 displays the effects of varying sampling Period $K$ on training a 7B-sized model using the 52K-entry Alpaca-GPT4 dataset. This graph contrasts loss curves for different sampling period $K$ values: $K = 122$ (green line), $K = 25$ (red line), and $K = 13$ (blue line) across 122 training steps. The results indicate that although each $K$ value results in distinct training trajectories, their convergence points are remarkably similar. This finding implies that for a 7B model

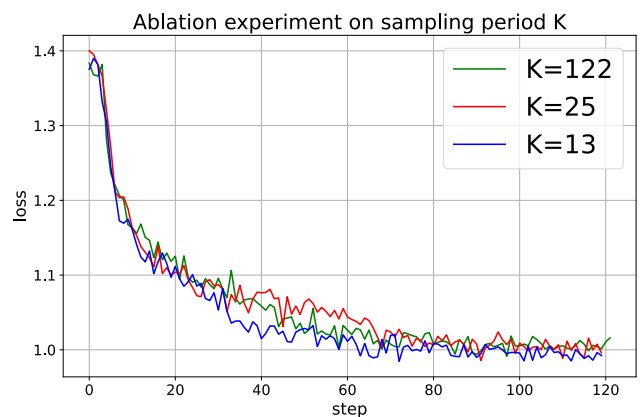

Figure 9: Comparison of loss curves for the sampling period $K$ ablation experiment.

trained on a 52K instruction conversation pair dataset, a sampling period of $K = 13$ is optimal for achieving the best loss curve and corresponding MT-Bench score radar graph.

### A.4.3 Sensitiveness to Randomness

LLaMA-2-7B on Alpaca-GPT4 with update step per sampling period $K = 13$, and sampling layers $\gamma = 2$, run three times with different random layer pick. Figure 10 shows that different random selections of layers slightly affect the training process but converge similarly. Despite initial fluctuations, the loss trends of three runs—distinguished by blue, green, and red lines—demonstrate that the model consistently reaches a stable state, underscoring the robustness of the training against the randomness in layer selection.

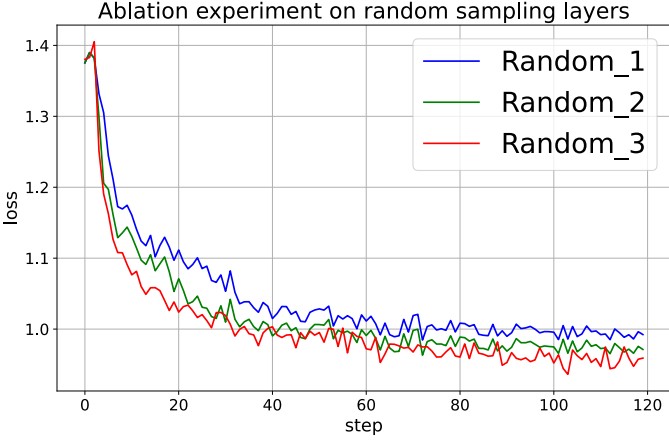

Figure 10: Comparison of loss curves for random variance ablation experiment, indicating the loss metric over steps.

Additionally, we also conduct experiments that analyze the impact of fixing randomly selected layers during training. The suggestion highlighted a need to evaluate the model's stability and performance under such conditions. To address this, we conducted further experiments using the LLaMA-2-7B model, maintaining identical hyperparameters and datasets as in our paper, and repeated the experiments three times to reduce variability from the random selection process.

Table 12: Compare LISA with fixed layers on LLaMA-2-7B, evaluate on MT-Bench.

| METHOD | MT-BENCH ↑ |
|---|---|
| LISA | **4.94** |
| LISA-FIX-1 | 4.62 |
| LISA-FIX-2 | 4.60 |
| LISA-FIX-3 | 4.67 |

The results are presented as table 12, with 'LISA-fix' denoting the experiments with randomly fixed layers and the appended number indicating the different selected seeds. The results are presented as table 12, with 'LISA-fix' denoting the experiments with randomly fixed layers and the appended number indicating the different selected seeds.

## A.5   Performance with Early Exiting

According to previous works [83, 84], the early exiting strategy in LLMs is effective. We are interested in investigating whether the LISA algorithm will have a different impact when combined with the early exiting strategy. To explore this, we conducted a series of experiments using the LLaMA-2-7B model, focusing on various training methods and early exit points. The early exiting method is DoLa [83].

Table 13: GSM8K Scores for LLaMA-2-7B when LISA meets the early exiting strategy DoLa.

| METHOD | GSM8K % ↑ |
|---|---|
| VANILLA | 15 |
| VANILLA + DOLA | 11 |
| FT + DOLA | 16 |
| **LISA** + DOLA | 17 |

We conducted all of our experiments using the LLaMA-2-7B model, selecting layers **[0, 8, 16, 32]** as early exit points for evaluation under four different training conditions: Vanilla (baseline), Vanilla with DoLa, Full Parameter Fine-Tuning (FT) with DoLa, and LISA with DoLa. The model used is the same one trained on the Alpaca-GPT4 dataset, as reported in section 4.2. This comprehensive approach enabled a detailed comparison of the model's performance and layer representation capabilities under various training methodologies. Due to time constraints, our experiments were conducted on a subset of 100 questions from the GSM8K test set, employing the same prompt as in the DoLa paper.

From the table 13, it can be seen that the LISA algorithm does not negatively affect the representation or performance of some layers of the model; instead, it contributes to some improvements in effectiveness.

## A.6   Comparison of Evaluation Loss

Besides the training loss, we also care about how LISA performs on the validation dataset. So, we split the Alpaca-GPT4 dataset into train and validation sets, the ratio is $9 : 1$, and ensure there is no data overlap between these two sets. Then we use the same setting in the sec 4.2, training the LLaMA-2-7B on the Alpaca-GPT4 dataset with full parameter training (FT), LoRA, GaLore, and LISA. As Figure 11 shows, the trend in validation loss mirrors that observed in training loss, with LISA exhibiting no signs of overfitting. This consistency underlines LISA's robustness in maintaining performance across different dataset splits.

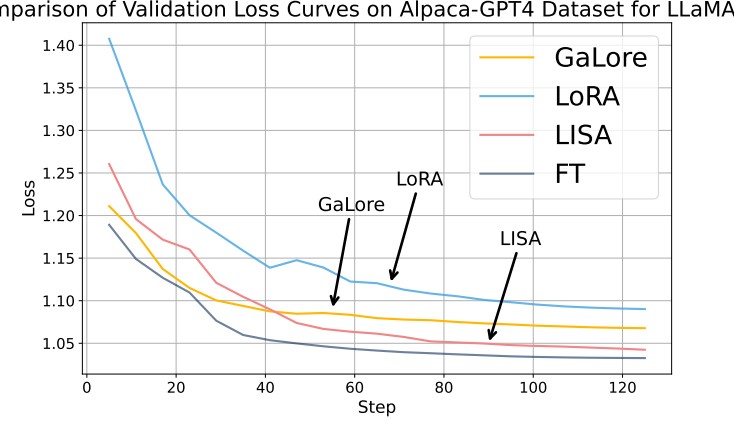

Figure 11: Validation loss comparison on the Alpaca-GPT4 dataset for LLaMA-2-7B, showing LISA, GaLore, LoRA, and FT strategies, with arrows indicating specific observations in the loss trends.

Table 15: The statistics of datasets. # TRAIN and # TEST denote the number of training and test samples respectively. The unit for OpenWebMath is the number of documents.

| DATASET | # TRAIN | # TEST |
|---|---|---|
| ALPACA GPT-4[60] | 52,000 | - |
| MT-BENCH [67] | - | 80 |
| GSM8K [73] | 7,473 | 1,319 |
| MMLU [68] | - | 14,079 |
| AGIEVAL [69] | - | 9316 |
| WINOGRANDE [70] | - | 44,000 |
| PUBMEDQA [74] | 211,269 | 1,000 |
| OPENWEBMATH [72] | 6.3M | - |

## A.7 Additional Observations of Layerwise Skewness

We conduct further weight norm experiments on Mistral-7 B to support our motivation that the bottom and top layers have a more significant impact on the output. Figure 12 provides similar observations as LLaMA-2-7B, where the bottom layer has a larger weight norm than other layers.

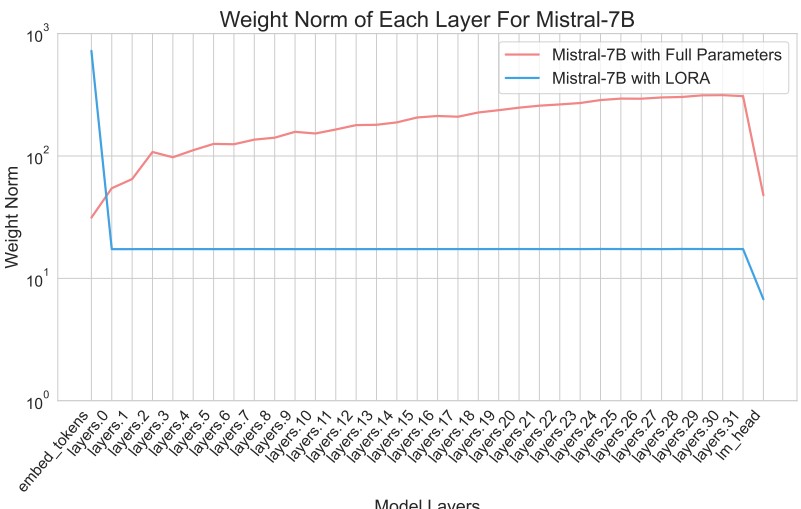

Figure 12: Layer-wise weight norms during training of Mistral-7B with LoRA and Full Parameters training.

# B  Training Setup and Hyperparameters

## B.1  Training Setup

In our experiments, we employ the LMFlow toolkit [85]* for conducting full parameter fine-tuning, LoRA tuning, and LISA tuning. We set the epoch number to 1 for fine-tuning and continual pre-training scenarios. Additionally, we utilized DeepSpeed offload technology [63] to run the LLMs

Table 14: Baseline Model Specifications

| MODEL NAME | # PARAMS | # LAYERS | MODEL DIM | # HEADS |
|---|---|---|---|---|
| TINYLLAMA | 1.1 B | 22 | 2048 | 32 |
| MISTRAL-7B | 7 B | 32 | 4096 | 32 |
| LLAMA-2-7B | 7 B | 32 | 4096 | 32 |
| LLAMA-2-70B | 70 B | 80 | 8192 | 64 |

efficiently. All experiments were conducted on $8\times$ NVIDIA Ampere Architecture GPU with 48 GB memory. Table 14 and table 15 are the information covered in this paper.

---

*https://github.com/OptimalScale/LMFlow

Table 16: The hyperparameter search identified optimal settings for each method: FP (Full Parameter Training), LoRA, GaLore, and LISA.

| | FP | LoRA | | LISA | | |
| --- | --- | --- | --- | --- | --- | --- |
| **Model** | lr | lr | Rank | lr | $\gamma$ | $K$ |
| GPT2-Small | $3 \times 10^{-4}$ | $6 \times 10^{-4}$ | 128 | $6 \times 10^{-4}$ | 2 | 3 |
| TinyLlama | $5 \times 10^{-6}$ | $5 \times 10^{-5}$ | 128 | $5 \times 10^{-5}$ | 2 | 10 |
| Mistral-7B | $5 \times 10^{-6}$ | $5 \times 10^{-5}$ | 128 | $5 \times 10^{-5}$ | 2 | 10 |
| LLaMA-2-7B | $5 \times 10^{-6}$ | $5 \times 10^{-5}$ | 128 | $5 \times 10^{-5}$ | 2 | 10 |
| LLaMA-2-70B | $5 \times 10^{-6}$ | $5 \times 10^{-5}$ | 128 | $5 \times 10^{-5}$ | 4 | 10 |

Our study explored a range of learning rates from $5 \times 10^{-6}$ to $3 \times 10^{-4}$, applying this spectrum to Full Parameter Training, LoRA, and LISA methods. For LoRA, we adjusted the rank $r$ to either 128 or 256 to vary the number of trainable parameters, applying LoRA across all linear layers. Regarding the number of sampling layers $\gamma$, our selections were guided by GPU memory considerations as reported in LoRA studies [9]; For the LISA algorithm, we selected $\gamma = 2$, and for experiments involving the 70B model, we opted for $\gamma = 4$. The sampling period ($K$), defined as the number of update steps per sampling interval, ranges from 1 to 50. This range was influenced by variables such as the size of the dataset, the batch size, and the number of training steps. To manage this effectively, we partitioned the entire training dataset into $K$ segments, thereby enabling precise regulation of the training steps within each sampling period.

## B.2 Continual Pre-training Dataset

We extracted a high-quality subset from OpenWebMath [72], using the '*Math_score*' attribute from the metadata as the metric for high-quality instances. The '*Math_Score*' represents the probability that a document is mathematical, and we set the threshold at 0.95. Finally, the number of tokens for this high-quality subset is 1.5 billion.

## B.3 Hyperparameter search

We commenced our study with a grid search covering (i) learning rate, (ii) number of sampling layers $\gamma$, and (iii) sampling period $K$. Noting the effective performance of the LoRA method, we set the rank value to $r = 128$ or $r = 256$.

The optimal learning rate was explored within the range $\{5 \times 10^{-6}, 10^{-5}, 5 \times 10^{-5}, 6 \times 10^{-4}, 3 \times 10^{-4}\}$, applicable to full parameter training, LoRA, and LISA. For GaLore, we adhered to the official Transformers implementation[*], utilizing default parameters, with the learning rate matching that of the full parameter training.

Regarding the number of sampling layers $\gamma$, in alignment with Table 1, we selected values that matched or were lower than LoRA's GPU memory cost. Consequently, $\gamma = 2$ was predominantly used in the LISA experiments, while $\gamma = 4$ was chosen for the 70B model experiments.

For the sampling period $K$, we examined values within $1, 3, 5, 10, 50, 80$, aiming to maintain the model's update steps within a range of 10 to 50 per sampling period. This selection was informed by dataset size, batch size, and total training steps.

The comprehensive results of our hyperparameter search, detailing the optimal values for each configuration, are presented in Table 16.

## C Licenses

For instruction following and domain-specific fine-tuning tasks, all the datasets, including Alpaca [61], GSM8k [73], MMLU [68], AGIEval [69] and PubMedQA [74] are released under MIT license.

---

[*]https://huggingface.co/blog/galore

WinoGrande [70] and MT-Bench [67] are under Apache-2.0 license. For GPT-4, the generated dataset is only for research purposes, which shall not violate its terms of use.

