# OpenReview forum: "LISA: Layerwise Importance Sampling for Memory-Efficient Large Language Model Fine-Tuning"
_NeurIPS.cc/2024/Conference — NeurIPS 2024 poster_

### Official Review · Reviewer_kABp · 2024-07-10

**Soundness:** 3
**Presentation:** 2
**Contribution:** 2
**Rating:** 5
**Confidence:** 4

**Summary:**

This paper proposes an parameter-efficient finetuning (PEFT) method, LISA, that only finetunes some sampled layers while freezing the rest  for some certain iterations, and resample the finetuned layers later.

To validate LISA's effectiveness, the authors benchmark it on various tasks, like instruction-following, medical QA, math problems, across various sizes of LLMs. The results show that LISA shares a similar GPU memory consumption as LoRA, 1.5x speedup than LoRA for training, better or comparable to LoRA and Full Finetuning for various tasks.

The main contributions of this paper are:
1. The proposed method is simple, easy to be applied;
2. The experiments are thorough and consistent to support LISA's effectiveness.

**Strengths:**

1. The paper is well-written and easy to follow;
2. The proposed method, LISA, is simple and practical;
3. The experiments are thorough, and the results are better than the baselines.

**Weaknesses:**

1. Lack of novelty. LISA is very similar to the method proposed in the paper [46] in the references. [46] proposes to pretrain a model layer-by-layer, while LISA shuffles the training order and finetune some for a period. Such novelty is not enough. In addition, the authors don't apply their finding, i.e. skewed weight-norm distribution across layers, to design a better sampling method. Instead, they apply uniform sampling, which makes the finding less relevant. BTW, the finding is very similar to a previous work [2] (Figure 4).

2. Lack of experimental details about the optimizer states, and unfair comparison. Compared to LoRA, LISA basically finetunes the whole model, which means the optimizer states stay the same as Full Finetuning. Since the oprimizer states occupy as twice memory as the model's, I assume LISA offload them to CPU. This part is not explicitely stated, nor in the limitation section. When comparing the training time, the authors only compare the forward and backward time (Figure 4), and ignore the latency introduced by the offloaded optimizer states, which is unfair since LoRA doesn't need to offload.

3. Lack of important ablation study. The benefit of LISA is from the training of the whole model in a layer-by-layer manner. Actually, we can also apply LISA to LoRA. I.e. we can use a very large size LoRA and only finetune the sampled LoRAs. In this way, LoRA has a larger number of trainable parameters, which might close the gap between LISA and LoRA.

4. Unintuitive results without further analysis. The most surprising result is that LISA outperforms full finetuning quite often, which is against our intuition. It also casts a doubt: how do you choose the hyper-parameters (like Table 15)? And what is the error bar for the main results.


[1] A fast learning algorithm for deep, Geoffrey E. Hinton, Simon Osindero, Yee-Whye Teh
[2] Parameter-Efficient Fine-Tuning without Introducing New Latency, Baohao Liao, Yan Meng, Christof Monz

**Questions:**

See weakness.

**Limitations:**

The authors discuss in the Limitation section.

---

> ### Author Rebuttal · Authors · 2024-08-06
>
> Thanks for the feedback. To address the raised concerns, we have provided additional clarifications and experiments, the details are listed as follows,
>
> **Weakness 1: Lack of novelty.**
> > LISA is very similar to the method proposed in the paper [46] in the references... In addition, the authors don't apply their finding, i.e. skewed weight-norm distribution across layers, to design a better sampling method. Instead, they apply uniform sampling, which makes the finding less relevant. BTW, the finding is very similar to a previous work [2] (Figure 4).
>
> Thanks for the valuable comments. We would like to emphasize that LISA's major novelty lies in its successful applications in LLMs, which significantly reduce memory consumption and improve the training speed of LLMs. In contrast, [46] is a paper in 2006 that focused on training of deep belief nets. Their settings are vastly different.
>
> Regarding the non-skewed weight-norm distribution claim, this may be a misunderstanding. The distribution in LISA is indeed skewed, as the embedding and linear head layers are always unfrozen during the training. We will add additional clarifications in Section 3.2 to make it clearer for readers.
>
> Concerning similarity to Figure 4 in [2], again, their settings are different, as the observations of [2] is made in encoder-only models, e.g. Roberta, while LISA focuses on decoder-only LLMs. In addition, we would like to emphasize again that the observation itself only serves as a motivation of our work, LISA's main novelty still lies in its successful applications in LLMs.
>
> **Weakness 2: Lack of experimental details about the optimizer states, and unfair comparison**
> > Compared to LoRA, LISA basically finetunes the whole model, which means the optimizer states stay the same as Full Finetuning. Since the oprimizer states occupy as twice memory as the model's, I assume LISA offload them to CPU. This part is not explicitely stated, nor in the limitation section. When comparing the training time, the authors only compare the forward and backward time (Figure 4), and ignore the latency introduced by the offloaded optimizer states, which is unfair since LoRA doesn't need to offload.
>
> Thanks for the feedback. We would like to kindly remind the reviewer that there may be some miscommunications, which lead to the false impressions here.
>
> In our implementation for all the experiments, the model will be reinitialized every time, so the optimizer is only applied to the trainable parameters and has the same or less memory consumption as LoRA, just as shown in Table 1 of the paper. Because of this, there is no need to offload the optimizer states to CPUs, let alone the latency of offloading optimizer states for LISA. We will emphasize these implementation details in our papers to avoid misunderstandings.
>
>
> **Weakness 3: LISA + LoRA experiment.**
> > The benefit of LISA is from the training of the whole model in a layer-by-layer manner. Actually, we can also apply LISA to LoRA. I.e. we can use a very large size LoRA and only finetune the sampled LoRAs. In this way, LoRA has a larger number of trainable parameters, which might close the gap between LISA and LoRA.
>
> Thank you for the suggestion. It is valid to combine LoRA with LISA. But if this approach works, logically it is just another proof of the effectiveness of LISA, not the opposite. Actually, LISA is orthogonal to most low-rank techniques, and there have already been published papers that combine LISA with GaLore [3]. Since the combination of those techniques already merits another paper, we intended to leave that part for future works.
>
> In addition, we have evidence showing that this simple combination does not work easily. As demonstrated in the table below, LoRA + LISA achieves similar performance as LoRA, but still significantly worse than LISA. The detailed results can be found in the attached rebuttal PDF.
>
> | Model          | Method   | MMLU (5-shot)      | AGIEval (3-shot)      | WinoGrande (5-shot)   |
> |----------------|----------|--------------------|-----------------------|-----------------------|
> | LLaMA-2-7B     | Vanilla  | 45.87              | 25.69                 | 74.11                 |
> |                | FT       | 45.66 ± 0.09       | **27.02 ± 0.10**      | 75.06 ± 0.13              |
> |                | LoRA     | 45.50 ± 0.07       | 24.73 ± 0.04          | 74.74 ± 0.09                  |
> |                | GaLore | 45.56 ± 0.05       | 24.39 ± 0.11          | 73.32 ± 0.12                |
> | | LISA + LoRA | 45.34 ± 0.41 | 25.55 ± 0.66 | 72.64 ± 0.24 |
> |                | LISA     | **46.21 ± 0.12**   | 26.06 ± 0.08          | **75.30 ± 0.11**                 |
>
> **Weakness 4: Unintuitive results.**
> > The most surprising result is that LISA outperforms full finetuning quite often, which is against our intuition. It also casts a doubt: how do you choose the hyper-parameters (like Table 15)? And what is the error bar for the main results.
>
> Thank you for the question. The intuition of this surprising effect can be attributed to the implicit regularization effect of LISA's layerwise freezing strategy, as mentioned in lines 245-250, where LISA and LoRA favor different types of tasks, implying the regularization tendency under different freezing strategies.
>
> The hyperparameter search process can be found in Appendix B.3, and we have conducted additional multi-trial experiments to address the raised concerns, please refer to the attached PDF for detailed results.
>
> > Reference:
> > [1] Geoffrey E. Hinton, Simon Osindero, and Yee-Whye Teh. A fast learning algorithm for deep belief nets. Neural Computation, page 1527–1554, Jul 2006.
> > [2] Liao, Baohao, Yan Meng, and Christof Monz. "Parameter-efficient fine-tuning without introducing new latency." arXiv preprint arXiv:2305.16742 (2023).
> > [3] Li, Pengxiang, et al. "OwLore: Outlier-weighed Layerwise Sampled Low-Rank Projection for Memory-Efficient LLM Fine-tuning." arXiv preprint arXiv:2405.18380 (2024).

---

> > ### Comment · Reviewer_kABp · 2024-08-13
> >
> > Thank you for the clarification and new eperiments. I'm willing to increase my score.

---

> > > ### Author Response · Authors · 2024-08-13
> > > **Thanks very much for your positive feedback!**
> > >
> > > Dear Reviewer kABp,
> > >
> > > We sincerely thank you for your positive feedback and the time you dedicated to reviewing our rebuttal. It brings us great joy to learn that our response has addressed your concerns and contributed to increasing the score from 4 to 5.
> > >
> > > As the score is still borderline, we are wondering if there are any major concerns regarding our current revision. It would be our great pleasure to provide further clarifications and results to address any additional doubts.
> > >
> > > Your suggestions really help a lot to improve our work and make the justification of our method more complete. We also greatly appreciate your recognition of the simplicity of our algorithm, the quality of our paper, and the contribution it makes.
> > >
> > > Once again, we would like to express our appreciation for your valuable comments during the reviewing process.
> > >
> > > Best regards,
> > >
> > > Authors.

---

> ### Author Response · Authors · 2024-08-09
> **Thank you very much for the valuable feedback!**
>
> Dear Reviewer kABp,
>
> Thank you very much for your valuable feedback! We truly hope that our new results can help clarify the concerns.
>
> We appreciate the time you have spent on reviewing our paper and providing us with the constructive comments. Please let us know if there are any concerns remaining. If you find our response to have addressed your concerns, it would mean very much to us if you could consider raising the score.

---

### Official Review · Reviewer_K468 · 2024-07-13

**Soundness:** 3
**Presentation:** 3
**Contribution:** 3
**Rating:** 7
**Confidence:** 3

**Summary:**

This paper proposes a lighter alternative to LoRA, LISA, based on using importance sampling to periodically choose a subset of layers to optimize. It is motivated by observations on the norm of parameter updates made during training with LoRA, compared to full parameter fine-tuning. Experiments are made comparing LISA with LoRA and full parameter fine-tuning on memory efficiency, and moderate/large fine-tuning and continual pre-training. Supplementary experiments are made on the hyperparameters of LISA and training sensitivity. Results show that LISA uses less memory than LoRA and achieves better results on most tasks. The paper also presents a discussion including theoretical properties of LISA.

**Strengths:**

- The method proposed by this paper is of obvious interest for efficient training.
- The experimental results presented are convincing, and seem to indicate that the method is promising.

**Weaknesses:**

- The main issue I have with this paper is that the approach presented does not seem to actually use Importance sampling: the weights of the $E$ and $H$ layers are fixed to $1.0$, and the intermediate layers have uniform weights. Besides, the presentation of the distribution and rates used in Section 3.2 is rather unclear.

**Questions:**

-  While the choice of that sampling distribution seems motivated by observation on LoRA, there seems to be shortcuts taken, particularly on fixing $E$ and $H$. Is there anything more to motivate this ? Did you try more extensive experiments with only $E + H$ ?
- Can you relate your approach with those mentioned in Section 2.2, focusing on layer-selection ?

**Limitations:**

- The limitations have been addressed.

---

> ### Author Rebuttal · Authors · 2024-08-06
>
> We would like to offer our sincere thanks for all your constructive comments and recognition of our contributions. We really appreciate it. Here we will address the raised concerns one by one.
>
> **Weakness: Sampling Strategy.**
> > The main issue I have with this paper is that the approach presented does not seem to actually use Importance sampling: the weights of the $E$ and $H$ layers are fixed to $1.0$, and the intermediate layers have uniform weights. Besides, the presentation of the distribution and rates used in Section 3.2 is rather unclear.
>
> Thank you for the valuable feedback! Importance sampling serves as the motivation of our algorithm, which emphasizes the difference between layers. To avoid misunderstandings, we will definitely clarify that in our revisions. Regarding the presentation issue in Section 3.2, we will add the description for different layers accordingly, such as $E$ and $H$, to make it clearer.
>
>
> **Question 1: Fixing $E + H$ layers.**
> > While the choice of that sampling distribution seems motivated by observation on LoRA, there seems to be shortcuts taken, particularly on fixing $E$ and $H$. Is there anything more to motivate this ? Did you try more extensive experiments with only $E+H$ ?
>
> Thank you for your insightful question! Yes. Besides the motivation we mentioned in the paper, we also found that fixing $E+H$ significantly hurts the performance, indicating the importance of those two layers.
>
> To further understand the effect of $E+H$ layers, we conducted additional experiments on LISA with only $E+H$ layers:
>
>
> | Model          | Method         | MMLU (5-shot) | AGIEval (3-shot) | WinoGrande (5-shot) | MT-Bench      |
> |----------------|----------------|---------------|------------------|---------------------|---------------|
> | TinyLlama      | Vanilla        | 25.50         | 19.55            | 59.91               | 1.25          |
> |                | FT             | 25.62 ± 0.10        | 21.28 ± 0.07           | **62.12 ± 0.15**              | 2.21 ± 0.16         |
> |                | LISA ($E+H$)   | 25.49 ± 0.14        | 20.75 ± 0.21           | 60.43 ± 0.19              | 2.18 ± 0.31             |
> |                | LISA           | **26.02 ± 0.13**        | **21.71 ± 0.09**           | 61.48 ± 0.08              | **2.57 ± 0.25**         |
> |||||||
> | Mistral-7B     | Vanilla        | 60.12         | 26.79            | 79.24               | 4.32          |
> |                | FT             | 61.70 ± 0.13        | 28.07 ± 0.12           | 78.85 ± 0.12              | 4.64 ± 0.12         |
> |                | LISA ($E+H$)   | 61.49 ± 0.12        | 27.66 ± 0.07           | 77.93 ± 0.11             | 4.51  ± 0.27            |
> |                | LISA           | **62.09 ± 0.10**        | **29.76 ± 0.09**            | **78.93 ± 0.08**              | **4.85 ± 0.14**           |
> |||||||
> | LLaMA-2-7B     | Vanilla        | 45.87         | 25.69            | 74.11               | 3.29          |
> |                | FT             | 45.66 ± 0.09        | **27.02 ± 0.10**            | 75.06 ± 0.13              | 4.75 ± 0.16         |
> |                | LISA ($E+H$)   | 45.88 ± 0.12        | 25.82 ± 0.15            | 73.48 ± 0.22              | 4.63 ± 0.35             |
> |                | LISA           | **46.21 ± 0.12**         | 26.06 ± 0.08           | **75.30 ± 0.11**              | **4.94 ± 0.14**        |
>
> As shown in the Table, LISA ($E+H$) can achieve reasonable performance but is still no match for LISA.
>
>
> **Question 2: Layer-selection related works.**
> > Can you relate your approach with those mentioned in Section 2.2, focusing on layer-selection ?
>
> Thanks for your constructive comments!
>
> Compared with the mentioned works, LISA's simple selection strategy makes it much easier to understand and implement, which reduces theoretical difficulties in analysis and engineering difficulties in practice.
>
> In contrast, Autofreeze [1] adopts a heuristic rule based on gradient norm changes to select layers, which is difficult to analyze in theory and not quite easy to control in practice. Autofreeze also focuses more on encoder-only models such as BERT, while LISA emphasizes its applications in LLMs.
>
> Freezeout [2] and SmartFrz [3] are facing the same issues as well, which progressively freeze layers, or adopt a NN-based predictor to guide the selection respectively. These heuristic-driven approaches are hard to obtain theoretical guarantees, and their empirical properties may vary as the backbone network changes. In addition, both Freezeout and SmartFrz focus on Computer Vision tasks, while LISA is adopted mostly in LLM settings.
>
> We will definitely include the aforementioned discussions in our revisions.
>
>
> > Reference
> > [1] Liu, Y., Agarwal, S., & Venkataraman, S. (2021). Autofreeze: Automatically freezing model blocks to accelerate fine-tuning. arXiv preprint arXiv:2102.01386.
> > [2] Brock, A., Lim, T., Ritchie, J. M., & Weston, N. (2017). Freezeout: Accelerate training by progressively freezing layers. arXiv preprint arXiv:1706.04983.
> > [3] Li, S., Yuan, G., Dai, Y., Zhang, Y., Wang, Y., & Tang, X. (2024). Smartfrz: An efficient training framework using attention-based layer freezing. arXiv preprint arXiv:2401.16720.

---

> > ### Comment · Reviewer_K468 · 2024-08-13
> >
> > I would like to thank the authors for their rebuttal, the clarifications and the supplementary results ! I have updated my *Soundness* rating, and maintain my global rating recommanding acceptance.

---

> > > ### Author Response · Authors · 2024-08-13
> > > **Thank you so much for your valuable feedback!**
> > >
> > > Dear Reviewer K468,
> > >
> > > We sincerely appreciate your time in reviewing our rebuttal and examining our new results. We are delighted to learn that our response successfully addressed your concerns and contributed to an increase in the _Soundness_ rating. Your constructive advice is really helpful in enhancing the completeness and quality of our paper!
> > >
> > > Once again, we sincerely appreciate your valuable feedback and consideration.
> > >
> > > Best regards,
> > >
> > > Authors

---

### Official Review · Reviewer_Rt8C · 2024-07-13

**Soundness:** 3
**Presentation:** 3
**Contribution:** 4
**Rating:** 7
**Confidence:** 4

**Summary:**

This paper proposes a new optimization algorithm called Layerwise Importance Sampled AdamW (LISA) for large language model (LLM) fine-tuning. The authors observe the skewed weight norms across different layers in the Low-Rank Adaptation (LoRA) method and use this observation to develop LISA, which randomly freezes certain layers during optimization. Experimental results show that LISA outperforms LoRA and full parameter training in various downstream fine-tuning tasks, demonstrating its effectiveness and memory efficiency.

**Strengths:**

(1) The paper addresses an important problem in the field of large language models, which is the high memory consumption during training. The proposed LISA algorithm provides a memory-efficient alternative to the existing LoRA method.

(2) The authors provide a clear motivation for their work and thoroughly analyze the layerwise properties of LoRA, which leads to the development of LISA. This analysis adds valuable insights to the field.

(3) The experimental results demonstrate the superiority of LISA over LoRA and full parameter training in various downstream tasks. The authors provide detailed comparisons and evaluations, supporting the claims made in the paper.

(4) The paper is generally well-written and easy to follow.

(5) The design of the proposed "Layerwise Importance Sampling AdamW" is quite interesting and novel, which may be generalized in other related LLM's variants.

**Weaknesses:**

(1) I am curious whether the authors will open-source their implementation? Since efficient training techniques are important to this field, it would be beneficial to have an easy-to-use implementation. In addition, it would be helpful if the authors provide more information on their hyperparameter tuning strategy in the experiments, like how do you choose the best hyperparameters?

(2) The some parts of the writing and layout needs improvement. For example, there are too many capitalized letters like line 6 "Parameter Efficient Fine-Tuning". I do not see any reason to capitalize every words. In addition, the figures should be re-organized like Figure 2.

(3) The authors seem to define 65B as a boundary for small language models and large language model (see line 56). Just curious, are there any justification or relevant claims? Since there are more and more small language model research nowadays and they are mainly focusing on 1B-7B, it would be interesting to have a clear definition in our community.

**Questions:**

None

**Limitations:**

The limitation is fine with me.

---

> ### Author Rebuttal · Authors · 2024-08-06
>
> We would like to offer our sincere thanks for all your constructive comments and recognition of our contributions. We really appreciate it. Here we will address the raised concerns one by one.
>
> **Weakness 1: Open-Source Implementation.**
> >  I am curious whether the authors will open-source their implementation? Since efficient training techniques are important to this field, it would be beneficial to have an easy-to-use implementation. In addition, it would be helpful if the authors provide more information on their hyperparameter tuning strategy in the experiments, like how do you choose the best hyperparameters?
>
> Thank you for your questions! The LISA algorithm has been integrated into several third-party libraries, such as LMFlow [1] and Axolotl [2], which have almost full support for single-GPU settings, and basic support for multi-GPU settings. The full support for multi-GPU settings will be available soon.
>
> For hyperparameters, we conducted extensive hyperparameter searches for learning rates for SFT, LoRA, and LISA, as well as for $K$ and $\gamma$ for LISA. Details can be found in Appendix B.3 of the paper.
>
>
> **Weakness 2: Writing and layout improvement.**
> >  The some parts of the writing and layout needs improvement. For example, there are too many capitalized letters like line 6 "Parameter Efficient Fine-Tuning". I do not see any reason to capitalize every words. In addition, the figures should be re-organized like Figure 2.
>
> We appreciate your feedback and will definitely improve these aspects in our revisions. Specifically, we will correct the unnecessary capitalization, such as in "Parameter Efficient Fine-Tuning," and reorganize the figures, including Figure 2, to enhance clarity and readability.
>
> **Weakness 3: Definition of small language models.**
> >  The authors seem to define 65B as a boundary for small language models and large language model (see line 56). Just curious, are there any justification or relevant claims? Since there are more and more small language model research nowadays and they are mainly focusing on 1B-7B, it would be interesting to have a clear definition in our community.
>
> In our experiment, we classify language models that can fit within a single 8xA40 (48GB) server as small language models ($\le$ 65B). This classification is based on practical considerations of model deployment and resource requirements. We will provide a more detailed description and clarify that in our revisions.
>
> By the way, to the best of our knowledge, 2B is normally the boundary for mobile LLMs, and 8B normally represents the boundary for single-GPU-trainable LLMs.
>
> > Reference:
> > [1] Shizhe Diao, Rui Pan, Hanze Dong, KaShun Shum, Jipeng Zhang, Wei Xiong, and Tong Zhang. 2024. LMFlow: An Extensible Toolkit for Finetuning and Inference of Large Foundation Models. In Proceedings of the 2024 Conference of the North American Chapter of the Association for Computational Linguistics: Human Language Technologies (Volume 3: System Demonstrations), pages 116–127, Mexico City, Mexico. Association for Computational Linguistics. https://github.com/OptimalScale/LMFlow
> > [2] Axolotl: https://github.com/axolotl-ai-cloud/axolotl

---

### Author Rebuttal · Authors · 2024-08-06

Thank you very much for all the constructive comments and suggestions! To further address every reviewer's concerns, we have included additional results in the attached PDF file. Please kindly refer to the file for more details.

---

> ### Author Response · Authors · 2024-08-12
> **Thanks for your valuable feedback!**
>
> Dear Reviewers,
>
> Thank you for your valuable feedback and for taking the time to review our submission. We appreciate your insights and have addressed your comments in our response. If there are any additional concerns or questions, please do not hesitate to reach out. We are more than happy to provide further clarification or discuss any points in more detail.
>
> Best regards,
>
> Authors

---

### Decision · Program_Chairs · 2024-09-25

**Decision:**

Accept (poster)

**Comment:**

The paper proposes a new PEFT method called LISA (Layerwise Importance Sampled AdamW). The authors observed that weight norms are skewed while fine-tuning the models. Inspired by the observation, the authors randomly freeze most middle layers during the optimization. The authors show that the proposed method uses similar or less memory than LoRA  while outperforming LoRA or full parameter fine-tuning in some cases.

Reasons to accept: All three reviewers have a consensus that the paper is well-organized and well-written. Also, they have a consensus that the problem to solve in the paper is important. Reviewer Rt8C and K468 mentioned that the experimental results supported the authors' claim well. Reviewer K468 had a question why E + H layers are handled differently, and the authors provided additional experiments why they are important. Reviewer kABp had concerns about novelty, unfair comparison and missing ablations. The authors clarified they were misunderstanding and provided additional information. Reviewer kABp increased the score based on the rebuttal.

Based on all those, the paper provides a more effective and more efficient way to fine-tune LLMs with supporting experiments which would be beneficial to the research community to enhancing the fine-tuning methods.